# Electrochemical Reduction and Voltammetric Sensing of Lindane at the Carbon (Glassy and Pencil) Electrodes

Nibedita Swain, Isha Soni, Pankaj Kumar and Gururaj Kudur Jayaprakash *

Laboratory of Quantum Electrochemistry, School of Advanced Chemical Sciences, Shoolini University, Bajhol, Solan 173229, Himachal Pradesh, India; swainnibedita97@gmail.com (N.S.); ishasoni065@gmail.com (I.S.); pankajsharma12j@gmail.com (P.K.)
* Correspondence: rajguru97@gmail.com

**Abstract:** In the agricultural field, pesticides are used tremendously to shield our crops from insects, weeds, and diseases. Only a small percentage of pesticides employed reach their intended target, and the remainder passes through the soil, contaminating ground and surface-water supplies, damaging the crop fields, and ultimately harming the crop, including humans and other creatures. Alternative approaches for pesticide measurement have recently received a lot of attention, thanks to the growing interest in the on-site detection of analytes using electrochemical techniques that can replace standard chromatographic procedures. Among all organochlorine pesticides such as gamma-lindane are hazardous, toxic, and omnipresent contaminants in the environment. Here, in this review, we summarize the different ways of the gamma-lindane detection, performing the electrochemical techniques viz cyclic, differential, square wave voltammetry, and amperometry using various bare and surface-modified glassy carbon and pencil carbon electrodes. The analytical performances are reported as the limit of detection 18.8 nM (GCE–AONP–PANI–SWCNT), 37,000 nM (GCE), 38.1 nM (Bare HBPE), 21.3 nM (Nyl-MHBPE); percentage recovery is 103%.

**Keywords:** gamma-lindane; electrochemical detection; glassy carbon electrode; pencil carbon

## 1. Introduction

Hexachlorocyclohexane (HCH) is an artificial organic pollutant also called hexachlorane. It has eight isomeric forms, but of these eight isomeric forms, four α, -β, -γ, and δ-HCHs are the most prevalent which is shown in Figure 1 [1]. From this compound, γ-HCH (also known as lindane) is the most constant and commonly used compound, and it is the supreme isomer. Lindane is a broad-spectrum chlorinated insecticide that has a mixture of several chemical forms of HCH and is written as γ-Hexachlorocyclohexane or γ-HCH. Organic pollutants are normally pesticides, insecticides, or fertilizer, but HCH is an insecticide that is used on fruits, plants, and animals. Lindane is one of the earliest generations of chlorinated organic insecticides, appearing shortly after the end of World War II [2]. All of the pollutants have the same physical and chemical properties. Therefore, it has also had PBT (Persistent, Bioaccumulative, and Toxic) properties. The reason is that it is long-lived, and another reason is the breakdown of the carbon chlorine bond is very slow, so it remains in the environment for a very long time. Additionally, having another property as a bioaccumulative chemical also means it acts as an endocrine disruptor. Lindane has been used as a widespread insecticide for seed, soil, tree, and wood treatments, cattle ectoparasite therapy, and human scabies and lice treatments in India and Canada [3].

Faraday was the first to synthesize the compound in 1825. Teunis van der Linden (1884–1965), a Dutch chemist, was the first to isolate and describe γ-hexachlorocyclohexane in 1912 [4]. These two scientists invested their work and knowledge in lindane to make it useful, but unfortunately for some chemical and physical properties, nations such as Albania, Argentina, Austria, Azerbaijan, Brazil, Bulgaria, China, Czech Republic, France,

Germany, Ghana, Hungary, India, Italy, Japan, Poland, Romania, Russia, Slovakia, Spain, Turkey, the United Kingdom, and the United States banned lindane manufacturing. However, some countries such as European nations as well as Asia and North America, began producing technical HCH and lindane. Overall, isomeric structure HCH has a high concentration of gamma-HCH: about 99%. It has the physical property of a solid, has a low vapor pressure, and is poorly soluble in water. However, it is quite soluble in organic solvents such as acetone, as well as aromatic and chlorinated solvents.

Voltammetric methods have many advantages over the other analytical methods. For example, we can analyze real samples and the redox process is also identified. It is a less time-consuming process and uses a portable instrument that is easy to handle and has good stability. For this reason, voltammetric sensing is used to detect lindane [5].

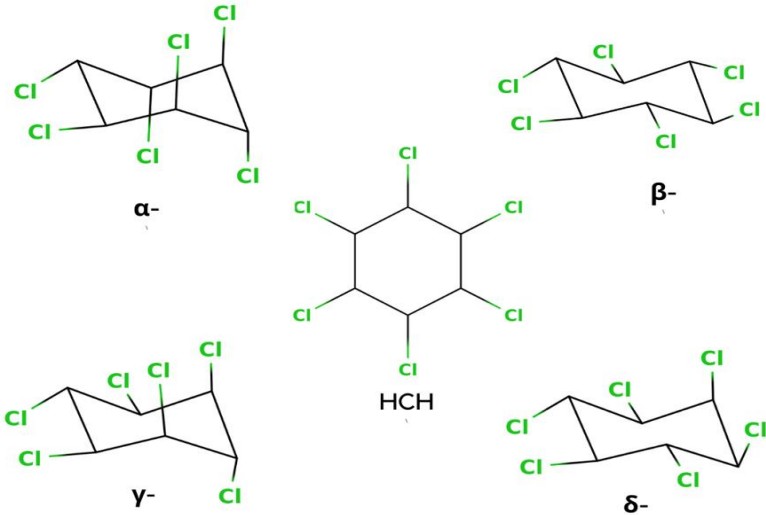

**Figure 1.** Isomeric Structures of HCH.

## 1.1. Environment Effect of Lindane

Lindane can be found in a variety of environments, with the majority found in water and the remainder in soil, sediment, and air [6]. The regions where lindane is created, used, or dumped are the most polluted. It can leak into groundwater, attach to soil particles, or evaporate into the atmosphere when present in the soil. The solubility of organic compounds in water and their inclination to attach to the soil, in general, regulate their leaching through the soil [7]. The carbon–chlorine bond is strong towards hydrolysis so the more chlorine is substituted, the more harm is caused to the biological and photolytic system. The breakdown of lindane is very slow and it remains in the environment for a long time. The presence of lindane affects the climate, global warming, as well as human and wildlife species [8].

Lindane also has a bioaccumulation effect, i.e., impacting vegetation and animals. Plants are also affected by the application of lindane for industrial, agricultural, or any kind of appliances. Some herbivorous mammals, such as cows and goats, are affected variously and the production of food and dairy is made hazardous by the use of lindane ($\gamma$-HCH) [9].

## 1.2. Human Health Effect

Lindane exposure in the general public can occur as a result of drinking water containing lindane, eating lindane-sprayed food, breathing lindane air mixture, or through other contacts [10,11]. Almost all human exposure to lindane comes from dietary sources. Lindane can be absorbed through the skin when scabies control lotions are used. Workers exposed to HCH through spraying have higher levels of lindane in their blood and skin lipids than the general population [12]. Lindane exposure in children is of particular concern; lindane has been discovered in maternal serum, placenta, and the umbilical cord.

In many parts of the world, HCH isomers have been discovered in human breast milk. Human infants may be exposed to lindane through breast milk [13]. People's cardiovascular, gastrointestinal, and reproductive systems are all affected by acute oral exposure. In humans, lindane's effect is acute and chronic; both possibilities are found [14].

### 1.3. Methods of Detection

There are several methods of detecting lindane ($\gamma$-HCH). Because of the toxicity of lindane, it is necessary to reduce or extract lindane from the worldwide environment. Many methods are used, such as Gas chromatography (GC), Liquid chromatography (LC), and multidimensional Gas chromatography (MDGC). An excellent potential separation method for the removal of lindane is high-performance liquid chromatography (HPLC). All of these techniques are expensive and need highly skilled laboratories [15].

Due to its low water solubility and strong negative reduction potential, electrochemical techniques of detection are not commonly accessible for lindane [16]. It has a high insecticidal behavior and has high reduction potential, so it can be detected by voltammetry analysis. Attempts to electrochemically reduce it to fully organic or mixtures of aqueous–organic media have been documented [17]. For the reduction, several electrode materials were used, including bare and modified carbon, silver, platinum, and copper. Lindane has redox qualities, so it will be more suitable for observing the electron transfer behavior in cyclic and square wave voltammetry [18], and certain electrodes are utilized to detect different other organic compounds, such as electron captured detection (EDC) and carbon-based electrode (CBE) [19], glassy carbon electrode (GCE), and screen-printed electrode (SPE) [20]. Of all of these methods, the most-employed technique is CBE, SPE, GCE, and HB pencil electrode.

Voltammetric techniques have several benefits over other detection methods [21,22], such as quantitative and qualitative determination of different species. They are easy to use and easy to handle. In the present review, we describe the electrochemical methods for the detection of lindane using carbon-based electrodes (glassy carbon and pencil graphite). These electrodes are easily available, cheaper, and beneficial to use many times.

## 2. Classification of Lindane Based on the Isomeric Form

HCH is a synthetic organochlorinated pollutant. There are eight isomeric versions of HCH. $\alpha$-, $\beta$-, $\gamma$-, and $\delta$-HCH are the four most universal isomers shown in Figure 2 [1]. $\gamma$-HCH (also known as lindane) is the most frequent isomer, and it is general. As the global usage of HCH decreases, so does the frequency of detection and the amounts identified in the environment [23].

In general, HCH formulation is a mixture of $\beta$- (5–12%), $\delta$- (6–10%), $\gamma$ (10–12%), and $\alpha$- (60–70%) isomers. However, only the $\gamma$-HCH isomer shows the maximum insecticidal activity. Therefore, it is used more in the agriculture sector than the other three isomers. $\gamma$-HCH is purified to at least 99% and the other isomers are discarded or not sent to the market [24].

HCH isomers are widely dispersed in the environment. They accumulate in the food chain and cause toxicity in biological systems due to their lipophilic characteristics. They are volatilized in the environment and transferred to far-flung locations. As a result, HCH isomers are among nature's most persistent and commonly encountered pollutants, with polluted sites documented all over the world. The half-life of lindane in soil and water was estimated to be 708 and 2292 days, respectively [25]. Lindane and other HCHs leftovers survive in the environment for a long time and have lately been found in water, soil, sediments, plants, and animals all over the world owing to their persistence [26].

**Figure 2.** Different stereochemical structures of lindane (α, β, γ, and δ).

### 2.1. Voltammetric Detection of Lindane Using Glassy Carbon Electrode

O.E. Fayemi et al. [27] prepared the environmentally friendly modified working electrodes by altering the surface of a glassy carbon electrode (GCE) using polyaniline nanofibers (PANI), nanoparticles (ZnO and $Fe_3O_4$), and multiwalled carbon nanotubes (MWCNTs). GCE is used as a material for electrodes in electrochemistry, as well as for high-temperature crucibles due to its thermostability, chemical resistance, and impermeability to gases and liquids. In the article, the authors prepared two types of composite electrodes (PANI/Zn, Fe(III) and Nylon 6,6/MWCNT/Zn, Fe(III)). Once preparing the electrodes, the authors used cyclic voltammetry (CV) to observe the reduction behavior of lindane. Both composite electrodes and bare GCE displayed irreversible (reduction) CV peaks, as shown in Figure 3. As shown in Figure 3, both the composite electrodes showed increased reduction peak currents for lindane electron-transfer reactions. GCE modified from Nylon 6,6/MWCNT/$Fe_3O$ showed the maximum sensing current because of improved diffusion of lindane molecules through the micropores of Nylon 6,6/MWCNT/MO nanofibers. They also performed the analysis of lindane in tap-water samples and obtained a recovery range performance.

A. Kumaravel et al. [28] prepared stable and reproducible cellulose acetate (CA)-modified GCE to sense lindane in the aqueous-alcoholic medium. The authors prepared the modified GCE using a drop-dry method (deposited CA solution on pretreated GCE). They have also used CV to test the electrode performance for sensing lindane. The authors observed the reduction peak of lindane at −1.5 V at both bare GCE and CA-modified GCE. However, at the CA-modified GCE, the reduction peak currents are quite high due to the CA-modified electron-transfer mechanism. They tested the analytical utility of the electrode in drinking water samples. They have used CV, differential pulse voltammetry (DPV), and amperometry to measure lindane in drinking-water samples. The recovery of lindane is higher at DPV.

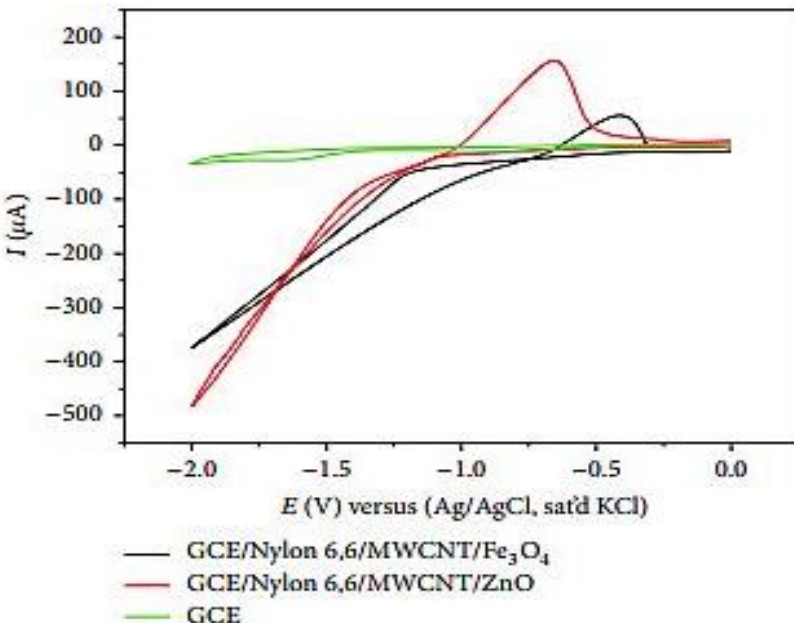

**Figure 3.** Comparative cyclic voltammograms for GCE, GCE/Nylon 6,6/MWCNT/ZnO, and GCE/Nylon 6,6/MWCNT/Fe$_3$O$_4$ at a scan rate of 50 mVs$^{-1}$ in 500 µM lindane in 60: 40 methanol/water containing 0.05 M TBAB. Different chemical structures of lindane (α, β, γ, and δ). Reprinted with permission from Hindawi Publishing Corporation, 2018 [27].

Anu Prathap et al. [2] detected lindane using NiCo$_2$O$_4$ modified glassy carbon electrode using differential pulse voltammetry (DPV) and cyclic voltammetry in the aqueous-methanol medium. At −1.5 V of the reduction peak, it shows a single well-defined irreversible lindane reduction. After the modification of the electrode, the reduction of lindane electron transfer can be calculated by the Randles–Sevcik equation. This modified GCE shows that lindane is irreversibly reduced, as there is no longer a reverse peak. They used four different metal oxides as an electrode, but all of these were found inactive to detect the reduction of lindane. Only NiCo2O4-modified GCEs are found to show a higher current response compared to other metal oxide electrodes. In this article, they have used CV, DPV, and amperometry techniques for the detection of lindane in tap water with good recovery in differential pulse voltammetry (DPV). The GCE–AONP–PANI–SWCNT modified electrode was used in this experiment, and square wave voltammograms were obtained with varied lindane concentrations. After the modification of GCE with NiCo$_2$O$_4$, it showed excellent electrolytic activity enhanced the electron transfer rate, and gave a detection limit of 37,000 nM. It was discovered that when the concentration of lindane increased, the lindane reduction peak current increased as well, and the potential gradually switched to the negative side [29]. The linear graph's standard deviation error value was found to be 0.21. The detection limit (LOD) for lindane at the GCE–AONP–PANI–SWCNT modified electrode was calculated using the relationship 3.3/m, where m is the slope of the same line and r is the relative standard deviation of the intercept of the y-coordinates. The GCE–AONP–PANI–SWCNT electrode's limit of detection (LOD), the limit of quantification (LOQ), and sensitivity to lindane were determined to be 2.01 nM, 6.09 nM, and 202.5 µA/µM, respectively, shown in Figure 4 [27]. This shows that the AONP–PANI–SWCNT-modified GCE has a superior electrocatalytic activity for lindane electrooxidation. The results showed that the GCE–AONP–PANI–SWCNT-modified electrode demonstrated anti-interference behavior when detecting lindane in the presence of interfering species (benzene, cyclohexane, phenol, Ca$^{2+}$, Fe$^{2+}$, K$^+$, and Mg$^{2+}$), with an average current drop of only 12.1 percent on the lindane reduction signal. To determine the practical application of the GCE–AONP–PANI–SWCNT-modified electrode for lindane detection in both river-water and tap-water samples, a real sample analysis was performed. Lindane/water samples with concentrations ranging from

0.5 to 100 M were generated using a 60:40 methanol/river-water mixture containing 0.05 M TBAB, and CV tests were conducted to measure the quantity of lindane in the samples. The recovery value of lindane at lower concentrations was found to be higher than the presence of lindane in river-water samples. The sensor's kinetic range could be responsible for this result. When comparing lower lindane concentrations to greater concentrations, the sensor produced good recoveries (99.9 percent average and 86.5 percent average) which is shown in Table 1. The lindane reduction mechanism proceeds with dissociative electron transfer (DET), which results in the scission of the carbon–chlorine bond. As illustrated in Scheme 1, the mechanism requires the production of an intermediate radical and an anion; Scheme 2 [30].

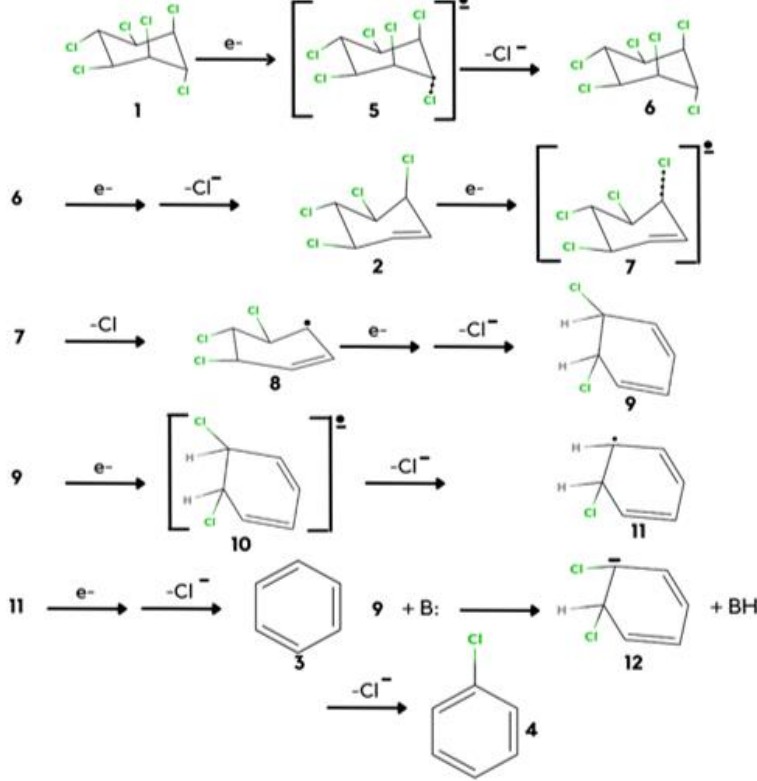

**Scheme 1.** Mechanism of lindane reduction by dissociative electron transfer leading scission of the carbon chlorine bond.

**Scheme 2.** The electrochemical reduction of hexachlorocyclohexanes.

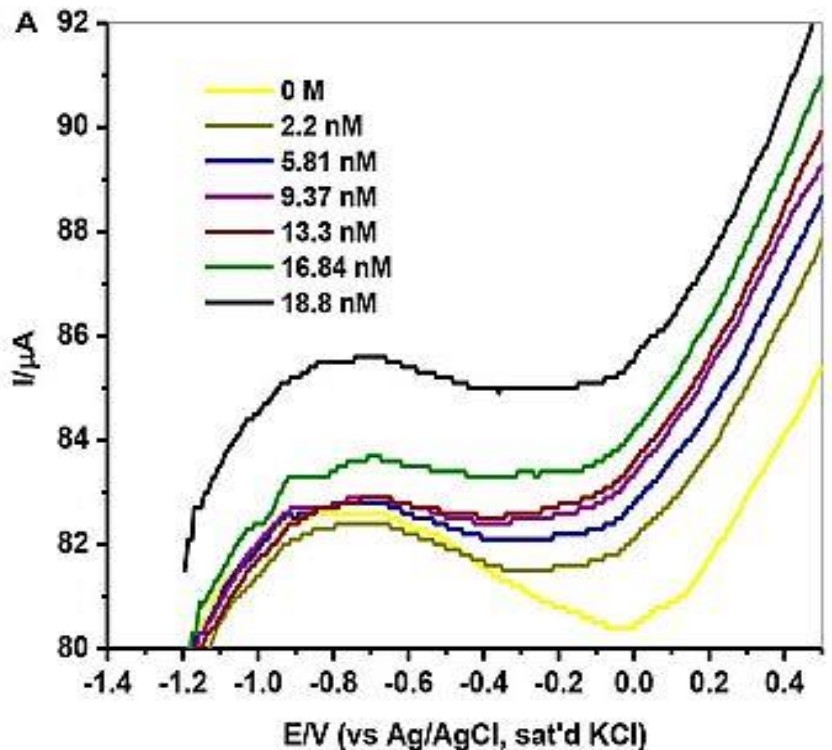

**Figure 4.** SWV of GCE–AONP–PANI–SWCNT electrode in 0.0–18.8 nM in lindane. Reprinted with permission from Hindawi Publishing Corporation, 2018 [27].

**Table 1.** Determination of lindane concentration in river water, tap water, lindane lotion, and drinking water using CV and SWV.

| S. No. | Sample | Modifier | LOD (nM) | Recovery (%) | Reference |
|---|---|---|---|---|---|
| 1 | River water | GCE–AONP–PANI–SWCNT | 18.8 | 0.35 | [27] |
| 2 | Tap water | GCE–AONP–PANI–SWCNT | 18.8 | 101 | [27] |
| 3 | Lotion | GCE | 37,000 | 87.5 | [28] |
| 4 | Drinking water | GCE | 37,000 | 87.57 | [28] |

### 2.2. Voltammetric Detection of Lindane Using Modified Pencil Carbon Electrode

Abdul Rahim Mohd Yusoff [29] proposed Nylon 6,6 modified graphite HB pencil as a working electrode to detect lindane using differential pulse cathodic stripping voltammetry (DPCSV). In this article, the author replaced the traditional mercury electrode to avoid the toxicity of mercury with economical HB pencil electrodes with Nylon 6,6 graphite HB pencil electrodes. Pencil graphite electrodes have received a lot of attention because of their low background currents, exceptional sensitivity, repeatability, implementable electroactive surface area, cost-effectiveness, and convenience of discharge. To modify the pencil electrode authors, use alumina paste, ethanol, and Nylon 6,6 solution. This modified electrode is proved that it has good stability, a good recovery percentage, and a low concentration could be determined. The recovery percentage of pesticides for water sample analysis was calculated using the following Equation (1).

$$\text{Recovery } (\%) = \frac{found\ concentration}{spond\ concentration} \times 100 \tag{1}$$

Some parameters are also used to detect lindane such as $E_i$ (effect of initial potential), $E_{ach}$ (effect of accumulation potential), and $T_{acc}$ (time accumulation) using the DPCSV technique. These parameters are best for the current peak of lindane using bare HBPE. The increasing and decreasing value of $E_{acc}$ (effect of initial accumulation) affects the peak value

of lindane. Additionally, the same trend can be observed by using Nyl-MHBPE. Therefore, the redox reaction of lindane is easier and it is more effective to detect the metals which are also present in the sample ($Cu^{2+}$, $Zn^{2+}$, $Pb^{2+}$, $Cd^{2+}$, and $Fe^{3+}$) [29].

At pH 10, the peak was shifted towards positive potential, but in the case of Nyl-MHBPE, it shifted towards the negative potential by increasing the pH value. The stripping signal at Nyl-MHBPE was greater as compared to bare HBPE.

To examine the lindane percentage at various concentrations, this effect of parameters helps to describe the electron chemical process in any dynamic potential techniques. For the analysis of lindane, the pH value should be eight for the bare HBPE and pH 7 for the Nyl-MHBPE. Modification of the electrode has very good sensitivity and shows the good electro-reduction of lindane because of the large surface area. The voltammogram analysis of lindane used bare HBPE and Nyl-MHBPE used the DPCSV technique. The results obtained for lindane determination show that the Nyl-MHBPE gives the lowest LOD value (21.3 nM) compared to the bare HBPE (38.1 nM). The recovery results of lindane were determined at both the bare HBPE and Nyl-MHBPE. For lindane determination, the recovery values of bare HBPE and Nyl-MHBPE are 95.3 to 99.4% with RSDs between 1.92 and 3.22 percent for river water, 98.7 to 103 percent with RSDs between 0.84 and 1.87 percent for drinking-water dispenser samples, and 97.8 to 103 percent with RSDs between 1.72 and 2.39 percent for tap-water samples confirms in Table 2. Notably, the Nyl-MHBPE had a higher sensitivity for lindane analysis than the plain HBPE. They determined lindane using the DPCSV method but also compared it with UV-vis spectrophotometry. They gave an overall overview after the examination of lindane that not much important difference was observed between Nyl-MHBPE and UV-vis spectrophotometry. Additionally, they also found that the detection limit (26.2 nM) was less than the other methods of LOD and it could be a disadvantage of the electrochemical method when comparing the spectrophotometry analysis. The DPCS voltammogram for lindane analysis at optimal experimental settings and both electrodes can be used to evaluate lindane electroactivity, although the Nyl-MHBPE provided a better stripping signal than the bare HBPE. The results also explain that Nyl-MHBPE shows high selectivity for lindane determination [30].

**Table 2.** Determination of lindane using water sample.

| S. No. | Sample | Modifier | LOD (nM) | Recovery (%) | Reference |
|--------|--------|----------|----------|--------------|-----------|
| 1 | River water | Bare HBPE | 38.1 | 99.41 | [29] |
| 2 | Tap water | Bare HBPE | 38.1 | 101.71 | [29] |
| 3 | Drinking water | Bare HBPE | 38.1 | 99.83 | [29] |
| 4 | River water | Nyl-MHBPE | 21.3 | 99.36 | [30] |
| 5 | Tap water | Nyl-MHBPE | 21.3 | 103.43 | [30] |
| 6 | Drinking water | Nyl-MHBPE | 21.3 | 103.43 | [30] |

### 2.3. Electrochemical Reduction of Lindane

There are a number of ways to reduce lindane, but if we focus on the electrochemical reduction, it has the property to pick out good oxidizing and reducing agents, so it is very useful to reduce lindane because it is a good reducing agent. The sequential electrochemical reduction of lindane takes place with one electron by expulsing chloride ions. As shown in Scheme 2.

STEP: 1

In this process weakness, the C-Cl bond cleaves and there will be the formation of an intermediate.

STEP: 2

Free radical ions will be formed in step 2 as a result of the remotion of the chlorine atom.

STEP: 3

The axial C-Cl bond is weakly held and completely broken by two hydrogen bonds. The chloride ion gives a conjugated intermediate.

STEP: 4

After the conjugated intermediate, the C- Cl bond cleaves and forms a radical ion compound.

STEP: 5

The radical ion forms a double bond that leads to the formation of a radical anion.

STEP: 6

Cl atom weakly binds to the benzene ring, the unusual structure forms, and due to the stability of the aromatic compound, the formations of benzene is the major product.

STEP: 7

The minor isomer can be produced from the same conformational isomer after the major product has been formed. The rupture of the C-Cl bond, which is held in a hydrogen-bonded structure, and removal of the chloride ion is given by comparing the whole six electron reduction of one to afford three. This method for the synthesis of 4 is less, so a base is required to approach and abstract an equatorial proton form. Chlorobenzene is a minor result of the reaction [31].

## 3. Conclusions

It is concluded that lindane is one of the most hazardous chemical moieties that acts as a pollutant. The detection and removal of lindane from real samples such as wastewater is challenging and has huge importance in analytical sciences because lindane has an undesirable effect on the environment. Therefore, it has become necessary to perform research on this and find an effective new method that does not affect it economically, and most importantly, it should be environment friendly. In particular, detection of lindane using voltammetric has several benefits such as the possibility of real sample analysis, less time taken, and a portable instrument. Based on our research survey, until now, very good research has been carried out to detect lindane in real samples using voltammetric methods such as CV and SWV.

However, to detect lindane it is necessary to modify the working electrode GCE using PANI, and SWCNT modifier has shown a significant advantage in detecting lindane in real samples with LOD 18.8 nM. Similarly, a pencil graphite electrode modified Nyl-MHBPE also showed promising applications in lindane detection at LOD 21.3 nM. The present short review gives ideas to readers on how to use nonmercury or environmentally friendly electrodes to detect lindane in real samples.

**Author Contributions:** Original draft preparation, resources, formal analysis, data curation, and visualization N.S.; formal analysis, resources, and data curation I.S. and P.K.; conceptualization, editing, and supervision, G.K.J. All authors have read and agreed to the published version of the manuscript.

**Funding:** This research received no external funding.

**Institutional Review Board Statement:** Not applicable.

**Informed Consent Statement:** Not applicable.

**Data Availability Statement:** Not applicable.

**Acknowledgments:** G.K.J. is thankful to B. E. Kumara Swamy, Kuvempu University, India, and also thankful to the Department of Science and Technology (DST) SERB TARE fellowship grant number TAR/2021/000197.

**Conflicts of Interest:** The authors declare no conflict of interest.

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
