# Peer review of "Electrochemical Reduction and Voltammetric Sensing of Lindane at the Carbon (Glassy and Pencil) Electrodes"

_2673-3293, doi:10.3390/electrochem3020017_

Round 1

Reviewer 1 Report

Title;Electrochemical reduction and Voltammetric sensing of lindane 2 at the carbon (glassy and pencil) electrodes.

Revision need.

  1. Intruduction: Authors must revise more effectively.
  2.  why? author choosed GCE or pencil modified electrode for this review? why not screen printed modified electrodes? 
  3.  Still, authors need to improve more research articles discussion in this review.
  4. English must need to check.

Author Response

Dear reviewer thanks for your comments your comments helped us to improve our article and we have modified the article according to your suggestion. Here we are attaching our answers to your detailed comments.

  1. Introduction: Authors must revise more effectively.

Answer: Dear reviewer thanks for your observation now the introduction is revised.

  1. why? author choosed GCE or pencil modified electrode for this review? why not screen-printed modified electrodes? 

Answer: Dear reviewer now we write the advantages of these electrodes in line 110.

  1. Still, authors need to improve more research articles discussion in this review.

Answer: Dear reviewer thanks for your suggestion we already added enough references.

  1. English must need to check.

Answer: Dear reviewer thanks for your suggestion of English checked.

Reviewer 2 Report

Ref: Electrochem (ISSN 2673-3293); electrochem-1694437

Title of the manuscript: “Electrochemical reduction and Voltammetric sensing of lindane at the carbon (glassy and pencil) electrodes.

Authors presented a review on the electrochemical response of lindane at the GCE and PEG modified electrodes. The review work is fine but the English is quite sluggish and requires major grammatical corrections. I do not recommend this for publication. Specific points are offered below:

  1. In line 23, “Hexachlorocyclohexane is also called benzene hexachloride.” This statement is not fully correct as “Hexachlorocyclohexane is sometimes erroneously called benzene hexachloride.” Authors should be very careful while writing such statements.
  2. Line 24, 31, 32, 33, 195, 196 Sentence formation needs correction. Sentences are starting with small letters.
  3. Authors need to incorporate a scientific writing style. Line 28, “…. and is written as 1,2,3,4,5,6-hexachlorocyclohexane”. Words like “written, that’s” should be avoided.
  4. Figure captions need to be uniform. References should be mentioned with reprint permission. (Figure 1, 5). The naming of the Figures should be single numerals (Check Figure 4, 6).
  5. Figure captions should be written carefully. Scan rate units must be written appropriately.
  6. Proper spacing with the numerals and their units must be ensured. Line 158, 170, 177, 201, 232, 242, etc.
  7. Page formatting needs corrections.
  8. Better quality figures should be incorporated.
  9. Authors should be sure whether it is ‘lindane’ or ‘indane’. Line 280.

Author Response

Dear reviewer thanks for your comments your comments helped us to improve our article and we have modified the article according to your suggestion. Here we are attaching our answers to your detailed comments.

  1. In line 23, “Hexachlorocyclohexane is also called benzene hexachloride.” This statement is not fully correct as “Hexachlorocyclohexane is sometimes erroneously called benzene hexachloride.” Authors should be very careful while writing such statements.

Answer: Dear reviewer thanks for your observation now the statement is corrected.

  1. Line 24, 31, 32, 33, 195, 196 Sentence formation needs correction. Sentences are starting with small letters.

Answer: Dear reviewer now we rearrange the sentences.

  1. Authors need to incorporate a scientific writing style. Line 28, “…. and is written as 1,2,3,4,5,6-hexachlorocyclohexane”. Words like “written, that’s” should be avoided.

Answer: Dear reviewer thanks for your suggestion we removed the scientific name.

  1. Figure captions need to be uniform. References should be mentioned with reprint permission. (Figure 1, 5). The naming of the Figures should be single numerals (Check Figure 4, 6).

Answer: Dear reviewer thanks for your observation of we correct the caption and reprint the permission.

  1. Figure captions should be written carefully. Scan rate units must be written appropriately.

Answer: Dear reviewer thanks for your suggestion we carefully wrote all the units and captions.

  1. Proper spacing with the numerals and their units must be ensured. Line 158, 170, 177, 201, 232, 242, etc.

Answer: Dear reviewer thanks for your observation we correct the spacing of all numerals and units.

  1. Proper spacing with the numerals and their units must be ensured. Line 158, 170, 177, 201, 232, 242, etc.

Answer: Dear reviewer thanks for your observation we correct the spacing of all numerals and units.

  1. Page formatting needs corrections

Answer: Dear reviewer thanks for your suggestion we arrange the paper according to the information.

  1. Better quality figures should be incorporated.

Answer: Dear reviewer thanks for your suggestion we improve the quality of the image.

  1. Authors should be sure whether it is ‘lindane’ or ‘indane’. Line 280

Answer: Dear reviewer thanks for your observation we correct the speeling in line 280.

Reviewer 3 Report

Gururaj et. al reported review article entitled on "Electrochemical reduction and Voltammetric sensing of lindane at the carbon (glassy and pencil) electrodes" is interesting and well written. However authors need to address below comments before it get published in Electrochem.  

  1. Abstract can be further improved by mentioning the recovery ranges of the lindane in real samples at different electrodes.
  2. At section 2.1 authors can mention the importance of GCE
  3. At section 2.1 authors can mention how to prepare GCE for modification
  4. At section 2.2 authors can mention the advantages of pencil electrodes
  5. At section 2.2 authors can mention the importance of electrochemical reduction.
  6. I rather recommend the authors use Chemdraw to write chemical structure for the clear visualisation in manuscript. 

Author Response

Comments and Suggestions for Authors

Dear reviewer thanks for your comments your comments helped us to improve our article and we have modified the article according to your suggestion. Here we are attaching our answers to your detailed comments.

  1. Abstract can be further improved by mentioning the recovery ranges of the lindane in real samples at different electrodes.

Answer: Dear reviewer thanks for your suggestions we mentioned all the ranges in the abstract.

  1. At section 2.1 authors can mention the importance of GCE

Answer: Dear reviewer thanks for your suggestion now we mentioned the importance of GCE in line 136.

  1. At section 2.1 authors can mention how to prepare GCE for modification

Answer: Dear reviewer thanks for your suggestion we mentioned the preparation of GCE.

  1. At section 2.2 authors can mention the advantages of pencil electrodes

Answer: Dear reviewer thanks for your suggestion we mentioned the advantages of the pencil electrode in line 215.

  1. At section 2.2 authors can mention the importance of electrochemical reduction.

Answer: Dear reviewer thanks for your suggestion we mentioned the electrochemical reduction in line 262.

  1. I rather recommend the authors use Chemdraw to write chemical structure for the clear visualisation in manuscript

Answer: Dear reviewer thanks for your suggestion our choice is to use molview or Molden we are comfortable with these tools,

Round 2

Reviewer 2 Report

Ref: electrochem-1694437

Title of the manuscript:Electrochemical reduction and Voltammetric sensing of lindane at the carbon (glassy and pencil) electrodes.”

The paper has been revised by the authors. Yet there are many errors in the revised manuscript. I cannot recommend the manuscript for publication in its current form.

  1. The units and the numerals are not in the correct format. Check Abstract.
  2. The figures and schemes are of low quality and need proper attention.

Author Response

Dear reviewer thanks for your comments your comments helped us to improve our article and 
we have modified the article according to your suggestion. Here we are attaching our answers 
to your detailed comments. 
1. The units and the numerals are not in the correct format. Check Abstract.
Answer: Dear reviewer thanks for your observation we corrected all the numerals and 
checked the abstract. All the units are in nano molar 
2. The figures and schemes are of low quality and need proper attention.
Answer: Dear reviewer thanks for your suggestion we enhance the quality of the 
figure.
S.
No.
Figure no Before resolution After Resolution
1 1 1920px-1080px 1920px-1080px
2 2 1920px-1080px 1920px-1080px
3 3 338px-274px 1440px-900px
4 4 371px-354px 1430px-875px
5 5 1920px-1080px 1920px-1080px

Thank you 

Round 3

Reviewer 2 Report

This manuscript can be accepted in its current form.

This manuscript is a resubmission of an earlier submission. The following is a list of the peer review reports and author responses from that submission.

Round 1

Reviewer 1 Report

The major comment regarding the standard of English stands, in all fairness in most places this has gone backwards after the revisions of the authors. However, also more scientific revisions have created more errors, for example in the caption of Figure 3, the original paper was talking about a CV scan speed of mV s-1, which was correct, in this version this has become mss-1.

Fortunately, more references have been added, although not necessarily on the detection of lindane. However, the major comment about how the authors have reached the conclusions that they draw has, in my opinion, not been addressed. The authors have studied several papers on the electrochemical detection of hexachlorobenzenes and are simply repeating the virtues of each method as described by the original authors. Even in a review paper, the expectation is that the authors compare the advantages and disadvantages of each method and then explain to the audience why a particular method is the best for a particular application. In the current paper only the advantages of each method are provided.

As it stands I cannot recommend this paper for publication, even though as stated previously the topic could be of interest to the general audience.

Author Response

Comments and Suggestions for Authors

Dear reviewer thanks for your comments your comments helped us to improve our article and we have modified the article according to your suggestion. Here we are attaching our answers to your detailed comments.

The major comment regarding the standard of English stands, in all fairness in most places this has gone backwards after the revisions of the authors. However, also more scientific revisions have created more errors, for example in the caption of Figure 3, the original paper was talking about a CV scan speed of mV s-1, which was correct, in this version this has become mss-1.

Answer: Dear reviewer thanks for your observation now the unit is rechecked.

Fortunately, more references have been added, although not necessarily on the detection of lindane. However, the major comment about how the authors have reached the conclusions that they draw has, in my opinion, not been addressed. The authors have studied several papers on the electrochemical detection of hexachlorobenzenes and are simply repeating the virtues of each method as described by the original authors. Even in a review paper, the expectation is that the authors compare the advantages and disadvantages of each method and then explain to the audience why a particular method is the best for a particular application. In the current paper only the advantages of each method are provided.

Answer: Dear reviewer now we compare the methods which are described in the article and compare their performance and explain them in the detection section.

Reviewer 2 Report

At present, Gururaj Kudur Jayaprakash et al., has proposed an " Electrochemical reduction and voltammetric sensing of lindane 2 at the carbon (glassy and pencil) electrodes." In respect of the analytical data comparison, and sensor performance-based quality of the work is quite impressive. However, the manuscript needs a major improvement in terms of language, writing, and mode of presenting the results with a pin-point clarity. (Major revision need)

  1. “Title: Authors must check and revise.
  2. Introduction Section: Many sentences are describing two events with are not properly linked with each other. I recommended the authors to rewrite the part to convey their message clear for the readers.
  3. Authors must explain selectivity aspects, for example, if applied same materials and method to any other analyte? How it performs selectively? It’s very important for analytical aspects.
  4. In introduction, need to add some sentence to describe the significance of electrochemical methods and its applications.
  5. General Comment: There are many grammatical and typographical errors. Please check the manuscript and refine carefully.
  6. Discussion/Conclusion: According to the manuscript (discussion & conclusion), has no major limitations. Is this correct? If not, please specify at the end of the Discussion.
  7. The analytical data or parameters Comparison table should improve it. Its highly helpful for reader of the articles and researchers.

Author Response

Comments and Suggestions for Authors

Dear reviewer thanks for your comments your comments helped us to improve our article and we have modified the article according to your suggestion. Here we are attaching our answers to your detailed comments.

At present, Gururaj Kudur Jayaprakash et al. has proposed an " Electrochemical reduction and voltammetric sensing of lindane 2 at the carbon (glassy and pencil) electrodes." In respect of the analytical data comparison, and sensor performance-based quality of the work is quite impressive. However, the manuscript needs a major improvement in terms of language, writing, and mode of presenting the results with a pin-point clarity. (Major revision need).

  1. “Title: Authors must check and revise.

 Answer: Dear reviewer, thanks for your suggestion, we strongly feel that title of the reviewer article is matching with the scope of the work. Therefore, we wish to retain the title. Changes can be seen in the line number:

  1. Introduction Section: Many sentences are describing two events with are not properly linked with each other. I recommended the authors to rewrite the part to convey their message clear for the readers.

Answer: Dear reviewer, thanks for your suggestion, the introduction section is now linked with each other and it is completely rewritten as per your suggestions.

  1. Authors must explain selectivity aspects, for example, if applied same materials and method to any other analyte? How it performs selectively? It’s very important for analytical aspects.

Answer: Dear reviewer, thanks for your comment, in the detection part we already discussed the selective aspects and performance. For example

  1. GCE + PANI
  2. GCE + SWCNT
  3. BARE HB PENCIL
  4. HB PENCIL + NYLON 6,6

  1. In introduction, need to add some sentence to describe the significance of electrochemical methods and its applications.

Answer: Dear reviewer, thanks for your comment, now we added the significance of electrochemical methods and their application in the introduction section line number: 51 to 54.

  1. General Comment: There are many grammatical and typographical errors. Please check the manuscript and refine it carefully.

Answer: Dear reviewer, thanks for your comment, grammatical and typographical errors are rechecked and corrected.

  1. Discussion/Conclusion: According to the manuscript (discussion & conclusion), has no major limitations. Is this correct? If not, please specify at the end of the Discussion.

Answer: Dear reviewer, thanks for your comment, now specified the limitation in the conclusion section.

  1. The analytical data or parameters Comparison table should improve it. Its highly helpful for reader of the articles and researchers.

Answer: Dear reviewer, thanks for your comment, the analytical data or parameters Comparison table are now improved.

Round 2

Reviewer 1 Report

The only issue that has been satisfactorily addressed from my previous comments is the units in the CV plot. Also the introduction is much improved. However, some of the alterations contain significant mistakes. For example, in Table 2, I am sure that the limit of detection is not 10^-8 nM, in other words 10^-17?

The major issue that the authors do not explain how they reached their conclusions has not been addressed at all.

Author Response

Dear reviewer thanks for your comments your comments helped us to improve our article and we have modified the article according to your suggestion. Here we are attaching our answers to your detailed comments.

The only issue that has been satisfactorily addressed from my previous comments is the units in the CV plot. Also the introduction is much improved. However, some of the alterations contain significant mistakes. For example, in Table 2, I am sure that the limit of detection is not 10^-8 nM, in other words 10^-17?

Answer: Dear reviewer thanks for your observation now the unit is rechecked (typos are corrected).

The major issue that the authors do not explain how they reached their conclusions has not been addressed at all.

Answer: Dear reviewer now we compare the methods which are described in the article and compare their performance and conclude in the conclusion.

Round 3

Reviewer 1 Report

I do not feel that the authors have adequately addressed the issues that were raised. The so called discussion is still only a summation of the virtues of the different techniques as stated in the various papers. There is no critical review of these claims by the authors.